# Geographical Disparity in Cardiorespiratory Fitness among 3,189,540 Japanese Children and Adolescents before and during the Coronavirus 2019 Pandemic: An Ecological Study

**DOI:** 10.3390/ijerph20075315

**Published:** 2023-03-29

**Authors:** Tetsuhiro Kidokoro

**Affiliations:** Research Institute for Health and Sport Science, Nippon Sport Science University, Tokyo 158-8508, Japan; kidokoro@nittai.ac.jp

**Keywords:** census, youth, physical fitness, inequalities, public health, temporal trends

## Abstract

This ecological study aimed to use nationally representative physical fitness (PF) data to investigate the geographical disparities in cardiorespiratory fitness (CRF) among Japanese children across prefectures before and during the coronavirus 2019 (COVID-19) pandemic. The publicly available descriptive PF data of children from Grade 5 (10–11 years; n = 1,946,437) and adolescents from Grade 8 (13–14 years; n = 1,243,103) at the prefecture level (47 prefectures) were obtained from the annual census PF survey in 2019 (before the pandemic) and 2021 (during the pandemic). The 20 m shuttle run performance was used as a measure of CRF. Geographical disparity was evaluated using the coefficient of variation (CV) for CRF across prefectures. There were significant negative relationships between the magnitude of infections (evaluated as the number of confirmed cases) and changes in CRF at the prefecture level (r ≤ −0.293, *p* < 0.05). This study also found a substantial increase in CVs of CRF across prefectures for Grade 8 students, suggesting that COVID-19-related restrictions had widened the geographical disparity in CRF among Japanese adolescents. Adolescents’ CRF is an important marker for current and future health; hence, the findings of widening geographical disparities in CRF are suggestive of widening geographical disparities in health among the Japanese population.

## 1. Introduction

The outbreak of the coronavirus disease in 2019 (COVID-19) has changed the way we live dramatically. Globally, pandemic-related restrictions (e.g., stay-home and social distancing measures, as well as temporary school closures) were implemented to control the number of people getting infected [1]. The accumulating evidence has shown that pandemic-related restrictions caused undesirable health behaviors (e.g., a decline in physical activity [PA], an increase in sedentary behaviors, and irregular sleep patterns) among school-aged children and adolescents [2,3]. Preliminary research has also shown that pandemic-related restrictions lead to declines in physical fitness (PF), including cardiorespiratory fitness (CRF), among children and adolescents [4,5,6,7,8,9]. For example, using the nationally representative PF data of 16,647,699 Japanese children and adolescents, we have previously found that the decline in CRF (evaluated from a 20 m shuttle run performance) during the pandemic (2019–2021) was 18-fold larger than the improvement seen before the pandemic (2013–2019) [9]. We also suggested that the corresponding declines in population health were due to the pandemic-related restrictions implemented in Japan [9] because CRF is significantly associated with current and future health [10,11,12].

Although all children across Japan experienced a series of pandemic-related restrictions (e.g., stay-home and cancellation and restrictions of sporting events and organized sports participation), the extent of restrictions differed based on the geographical area of residence. Overall, children who resided in areas with the highest number of infections encountered maximum movement restrictions [13,14,15]. This is because local governments (i.e., prefectural governors), instead of central governments, played an important role in controlling the number of infected individuals based on their local conditions (e.g., the number of infected people in the prefecture) [13,14]. For example, even after the first state of emergency was lifted in the majority of the prefectures (on 14 May 2020), the emergency was extended in the most affected prefectures (i.e., Hokkaido, Tokyo, Kanagawa, Saitama, Chiba, Osaka, Kyoto, and Hyogo) [14]. Given the different reactions to the pandemic across the prefectures, it is possible that children and adolescents who resided in the most affected prefectures were more likely to be affected by pandemic-related restrictions. However, to date, no study has examined the association between the extent of infection and the changes in CRF in children and adolescents. 

In Japan, a census PF survey (hereafter called the JP Fit Survey for Youth) has been conducted annually for Grade 5 (aged 10–11 years) and Grade 8 (aged 13–14 years) students since 2008, and nearly all students in the age groups undergo the PF testing annually (approximately 2 million per annum) [16]. The descriptive results, including the CRF, were reported by the Japan Sport Agency. In addition, descriptive data at prefecture levels (47 prefectures) are publicly available, which enables us to understand the changes in CRF at prefecture levels during the pandemic [16]. Using the nationally representative PF data, this ecological study aimed to investigate the geographical disparity in CRF among Japanese children at the prefecture level before and during the pandemic. 

## 2. Materials and Methods

### 2.1. Japanese PF Surveillance

This study used nationally representative CRF data from the JP Fit Survey for Youth [16]. Publicly available descriptive PF data of children in Grade 5 (10–11 years; n = 1,946,437) and adolescents in Grade 8 (13–14 years; n = 1,243,103) in 47 prefectures were obtained from the annual census PF survey in 2019 (before the pandemic) and 2021 (during the pandemic). The PF participation rates at the school level in 2019 were 98.0% and 95.5% for Grade 5 and 8 students, respectively. Meanwhile, the participation rates in 2021 were 97.6% and 95.3%, for Grade 5 and 8 students, respectively. In both the years (2019 and 2021) the PF tests were administered between May and July. 

CRF was evaluated in schools based on the results of a 20 m shuttle run; this test is commonly conducted by classroom or physical education teachers according to standardized protocols [17,18]. In this study, the participants were asked to run back and forth over 20 m while the running pace was progressively increased. The participants were asked to continue running until fatigued (i.e., failure to keep up with the running speed twice). The highest number of running laps that each participant completed in the test trial was recorded as their test performance. To examine the changes in CRF during the COVID-19 pandemic, data from the pre-pandemic (CRF in 2019) and during the pandemic (CRF in 2021) were collected. The 2021 data were used because the 2020 PF survey was cancelled due to the widespread COVID-19 pandemic. 

### 2.2. Self-Reported Exercise Time

The PF survey involved a questionnaire that was administered in schools by classroom or physical education teachers. Self-reported exercise time was quantified using the following question: “Usually, for how long do you play sports or exercise besides the physical education classes”? Participants were asked to quantify the time spent exercising for all days of the week (Monday through Sunday). The total exercise time was calculated, and average daily exercise time was calculated by dividing the total time by seven. Self-reported time spent exercising in an organized sports club was also quantified in the questionnaire, although this question was limited to only Grade 8 students. Participants were asked whether they engaged in any of the following activities (multiple choice): (1) organized sports club at school, (2) non-sports club at school, and (3) sports club in the community, or if they (4) were not involved in any club activity. For those who belonged to an organized sports club at school, the following question was asked: “Usually, how long do you exercise in the organized sports club”? Participants were asked to quantify the time spent exercising in organized sports clubs for all days of the week (Monday through Sunday), and the total exercise time in the organized sports club was calculated. The average daily exercise time was calculated by dividing the total time by seven. 

### 2.3. Data Analyses

Descriptive results of CRF (e.g., sample sizes, means, and standard deviation (SD)) at prefecture levels were reported annually and used in this ecological study (47 prefectures). Only the mean values were available for the variables obtained by the questionnaire (self-reported exercise time and exercise time in organized sports at school). Geographical disparity in CRF across prefectures was quantified as the coefficient of variation (CV), which is the ratio of the SD to the mean. Trends in CVs were examined by calculating the ratio obtained by dividing the 2021 CV by the 2019 CV [19,20]. Ratios >1.1 indicated substantial increase in variability (i.e., the magnitude of variability in relation to the increase in mean over time); ratios < 0.9 indicated substantial decline in variability (i.e., the magnitude of variability in relation to the decrease in mean over time); and ratios between 0.9 and 1.1 indicated negligible trends in variability (i.e., the magnitude of variability in relation to the mean that did not change substantially over time) [21]. Standardized (Cohen’s) effect sizes (ES) were estimated to examine the magnitude of change in CRF [22]. A positive ES indicated improvement in means, while a negative ES indicated a decline in means.

This study also examined the associations between the cumulative number of COVID-19 cases at the prefecture level and the changes in mean CRF using Pearson correlation coefficients (r). Additionally, this study examined the associations among changes in exercise time, changes in exercise time in organized sports clubs, and the cumulative number of COVID-19 cases at prefecture levels. To interpret the magnitude of correlation, r values of 0.3, 0.5, and 0.7 were used for weak, moderate, and strong correlations, respectively, with correlations <0.3 considered to be negligible [23]. The cumulative number of COVID-19 cases at the prefecture level was reported by the Ministry of Health, Labour, and Welfare of Japan [24]. The cumulative number of cases at prefecture level by 30 April 2021, was used, as the PF test was administered from 1 May 2021.

## 3. Results

Figure 1 shows the cumulative number of newly confirmed cases per 100,000 people in the prefecture (by 30 April 2021). There were substantial differences in the cumulative confirmed cases across the prefectures. Overall, the confirmed cases were the largest in metropolitan cities, such as Tokyo and Osaka and their sounded prefectures (i.e., the Kanto and Kansai region). Additionally, the Okinawa prefecture (southernmost and westernmost prefectures) had the third highest number of cases after Tokyo and Osaka. In contrast, the cases were relatively smaller in the north of Japan (Tohoku region, such as Akita and Iwate) as well as in the westernmost region of the main island (Chugoku region, such as Shimane and Tottori; Figure 1).

Table 1 shows the descriptive characteristics of CRF and the exercise times in 2019 and 2021. The CRF declined from 2019 to 2021 for all age–sex groups. Specifically, CRF declined by 6.9%, 6.5%, 4.4%, and 7.0% for Grade 5 boys, Grade 5 girls, Grade 8 boys, and Grade 8 girls, respectively. The self-reported exercise time also declined from 2019 to 2021 for all the age–sex groups. Specifically, the self-reported exercise time decreased by 6.8%, 4.8%, 13.2%, and 15.0% for Grade 5 boys, Grade 5 girls, Grade 8 boys, and Grade 8 girls, respectively. Furthermore, the self-reported exercise time in organized sport clubs (only for Grade 8 students) decreased during this period. Specifically, 17.9% and 20.0% decreases in the exercise time in organized sport clubs were found for boys and girls, respectively. 

Figure 2 shows the CRF landscapes for Grade 5 students in 2019 and 2021. In 2019, CRF was higher in the Hokuriku region (e.g., Fukui, Ishikawa, and Niigata prefectures) and Tohoku region (Akita prefecture) compared to that in the Tokai region (e.g., Aichi prefecture), Hokkaido prefecture, and Okinawa prefecture. In 2021, the CRF in almost all the prefectures substantially declined compared with that in 2019, for both boys and girls.

Figure 3 shows the CRF landscapes for Grade 8 students in 2019 and 2021. In 2019, CRF was higher in the Fukui, Nagasaki, Saitama, Miyazaki, and Ibaraki prefectures, while CRF was lower in the Okinawa, Hokkaido, Wakayama, Kochi, and Tokyo prefectures. In 2021, the CRF in almost all the prefectures substantially declined compared with that in 2019, for both boys and girls.

Figure 4 shows the changes in CRF (expressed as ES) according to grade and sex between 2019 and 2021. There were substantial differences in the change in CRF between 2019 and 2021. Overall, the largest declines were found in the Kanto region (for both Grade 5 and Grade 8 students), Tokai region (for Grade 8 students), and the Kansai region (for Grade 5 girls). 

Figure 5 shows the changes in the variability of the CRF at the prefecture level between 2019 and 2021. For Grade 8 students (both boys and girls), the variability of the CRF substantially increased in 2021 compared to 2019 (i.e., the ratio of CVs for Grade 8 boys and girls was 1.29 and 1.20, respectively). In contrast, the changes in variability for Grade 5 students (both boys and girls) were negligible (i.e., the ratio of CVs for Grade 5 boys and girls was 1.03 and 1.00, respectively). 

Figure 6 shows the associations between the cumulative confirmed cases and changes in CRF at the prefecture level. For Grade 5 students, there was a negligible to weak association between the cumulative confirmed cases and the changes in CRF for boys (r = −0.293, *p* = 0.045) and girls (r = −0.316, *p* = 0.030). For Grade 8 students, there was a moderate association between the cumulative confirmed cases and the changes in CRF for boys (r = −0.553, *p* < 0.001) and girls (r = −0.655, *p* < 0.001).

Table 2 shows the association among the change in exercise time, the change in CRF, and the number of confirmed cases at the prefecture level according to the grade and sex group. For Grade 5 students (both boys and girls), there were no significant associations between the change in exercise time, the change in CRF, and the number of confirmed cases. In contrast, there were significant associations between the change in exercise time and the change in CRF among Grade 8 students (both boys and girls, r ≥ 0.440). Additionally, the changes in exercise time in organized sports clubs were significantly associated with changes in CRF (r ≥ 0.461). In contrast, the number of confirmed cases was negatively associated with change in exercise time (boys: r = −0.615, girls: r = −0.621, *p* for both < 0.001) and change in exercise time in organized sports clubs (boys: r = −0.691, girls: r = −0.704, *p* for both < 0.001) for Grade 8 students. 

## 4. Discussion

This ecological study examined the changes in the geographical disparities in CRF among Japanese children before and during the COVID-19 pandemic. Compared to before the pandemic (i.e., in 2019), the CRF levels substantially decreased in almost all prefectures during the pandemic (i.e., in 2021), with the largest declines observed in the most affected prefectures. This study also found substantial increases in the variability of the CRF across prefectures for Grade 8 students, suggesting that COVID-19-related restrictions widened the geographical disparity in the CRF among Japanese adolescents. In contrast, the changes in the CRF variability for Grade 5 students during the pandemic were negligible. These results suggest that age- and geography-specific approaches should be implemented to help Japanese children and adolescents recapture their pre-pandemic PF levels.

### 4.1. Substantial Variations in the Magnitude of COVID-19-Related Restrictions and the Changes in CRF at Prefecture Levels

Although the accumulating evidence showed that COVID-19-related restrictions have negatively affected children’s PF, including CRF [4,5,6,7,8,9], to date, no study has examined the changes in geographical disparity in CRF among children and adolescents during the pandemic. Importantly, there were substantial differences in the magnitude of COVID-19-related restrictions across the prefectures, depending on the local infection status [13,14,15]. This ecological study was the first to demonstrate significant relationships between the magnitude of infections (evaluated as the number of confirmed cases) and changes in CRF at the prefecture level. These results suggest that implementing stricter COVID-19 restrictions is associated with a larger decline in CRF. Although CRF in children is known to be influenced by numerous physiological and psychological factors [25,26], the main driver of CRF is the time spent engaging in PA, particularly the endurance types of PA [27]. Therefore, international experts have recommended the use of CRF as an indicator of recent PA status [28,29]. Given that children’s CRF is an important marker for current and future health [10,11,12], the findings of widening geographical disparities in CRF are suggestive of widening geographical disparities in health among the Japanese population.

### 4.2. Different Associations by Age Groups

The present study showed that the associations between the magnitude of infections and changes in CRF were stronger in Grade 8 students than in Grade 5 students. While it is not clear why this is the case, cancellations and/or restrictions of organized school sports activities may explain the different associations according to age groups. In Japan, organized school sports within school settings are a major source of PA opportunities, particularly for junior high school students (Grades 7–9), and the majority of adolescents participate in organized sports at school. Indeed, a national representative survey showed that 74.0% and 49.8% of junior high school boys and girls, respectively, participated in at least one sports club at school, and the participation rate was higher than that of elementary school students (Grades 1–6) [30]. However, it was reported that the frequency and duration of organized sports were significantly reduced after the outbreak of the COVID-19 pandemic [30]. This may have deprived students of PA opportunities, particularly junior high school students. In line with this hypothesis, the present study found that changes in time spent in organized sports clubs were positively associated with changes in CRF and negatively associated with the magnitude of infections in the prefecture. In contrast, there was no significant association between the changes in exercise time and CRF among Grade 5 students (both boys and girls). Given that only the exercise time, not PA, was self-reported in the JP Fit Survey for Youth, and that active play is a major source of PA opportunities for Japanese elementary school children [30], it is possible that the time spent in active play among Japanese elementary school children declined during the pandemic. Since children’s active play is known to be influenced by their surrounding environments [31,32], it is possible that the pandemic-related restrictions may have impacted the quantity and intensity of active play, resulting in a decline in CRF among elementary school students. These differences in the PA patterns between elementary and junior high school students may explain the different associations according to age group.

### 4.3. Strength and Limitation

The present study has several strengths. First, nationally representative PF data from the JP Fit Survey for Youth was used, in which nearly all Japanese students in Grades 5 and 8 participated (the school level participation rates were >95%). Therefore, the census data allowed us to understand the complete national picture of PF in Japanese children and adolescents. Second, using the cumulative number of COVID-19 cases reported by our government, a significant association was found between the magnitude of infections and the changes in CRF, which is unique to the literature. The findings of this study may support the notion that implementing stricter restrictions leads to a greater decline in CRF among children and adolescents. However, this study had some limitations. First, the data used in this ecological study are based on prefecture averages (as opposed to individual data); therefore, it remains unclear whether the same results can be confirmed in studies using individual data. Second, the exercise time in the present study was self-reported, which may have been inaccurate. However, it is important to note that the JP Fit Survey for Youth used the same question for exercise time for both years (2019 and 2021); therefore, the results from this study were less likely to be biased. 

## 5. Conclusions

The present study found that there were significant associations between the cumulative COVID-19 confirmed cases and the magnitude of changes in CRF in the prefectures, suggesting that CRF in children and adolescents living in the most infected areas was the most affected. Additionally, there was a substantial increase in the variability of the CRF across the prefectures among Japanese adolescents. Given that the CRF of adolescents is an important marker for current and future health, the findings of widening geographical disparities in CRF are suggestive of widening geographical disparities in health among the Japanese population. This study calls for age- and geography-specific approaches to help Japanese children and adolescents recapture their pre-pandemic PF levels.

## Figures and Tables

**Figure 1 ijerph-20-05315-f001:**
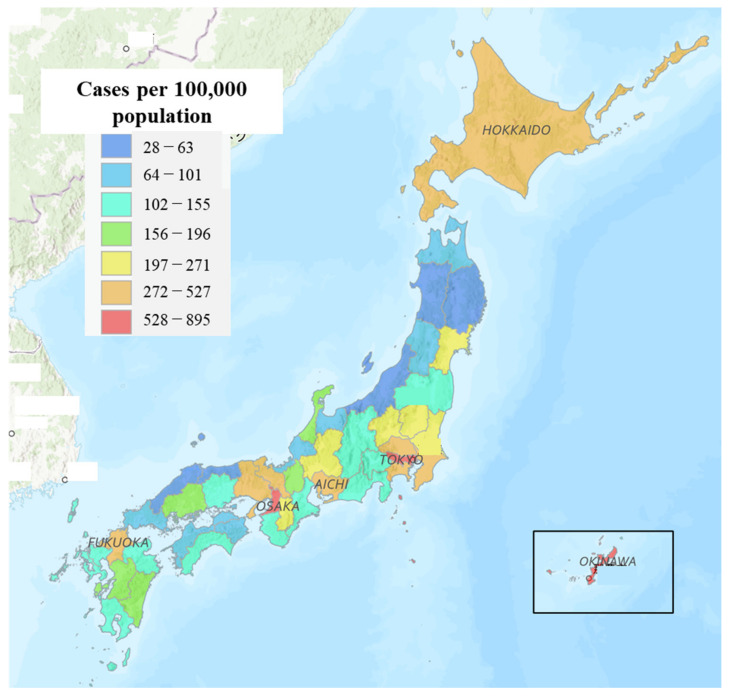
Cumulative newly confirmed cases per 100,000 people at various prefectures.

**Figure 2 ijerph-20-05315-f002:**
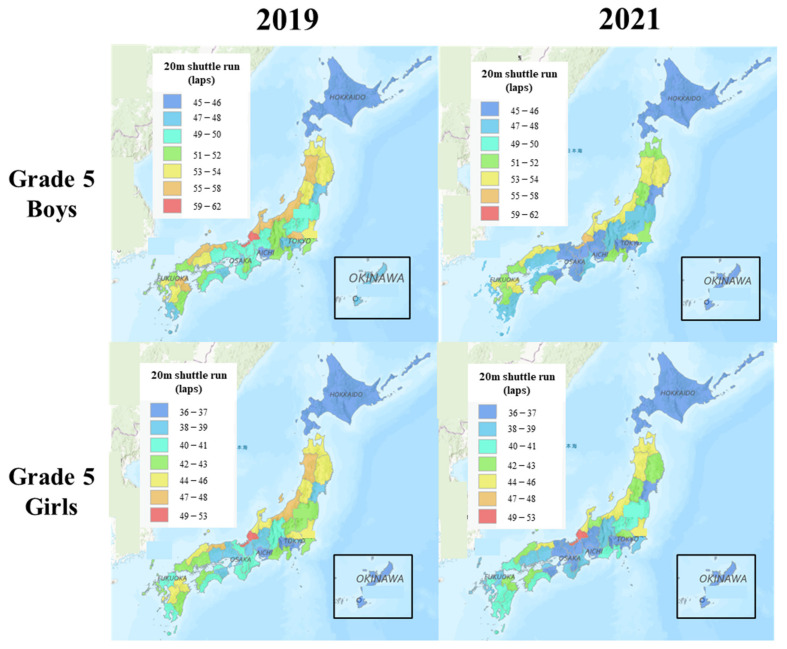
CRF landscapes for Grade 5 students in 2019 and 2021.

**Figure 3 ijerph-20-05315-f003:**
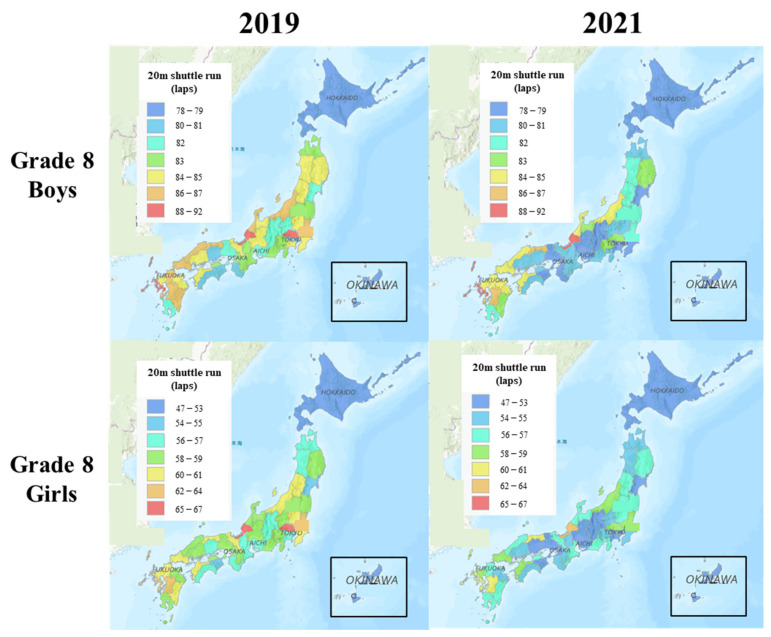
CRF landscapes for Grade 8 students in 2019 and 2021.

**Figure 4 ijerph-20-05315-f004:**
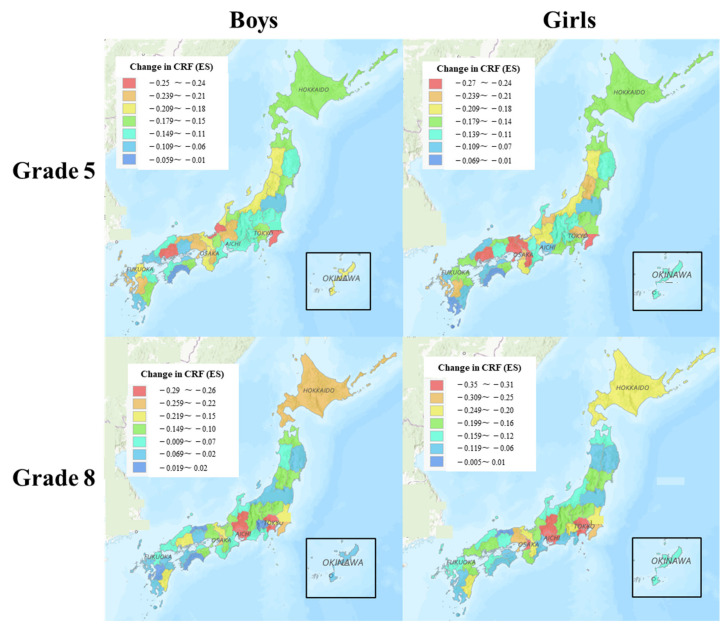
Changes in CRF according to Grade and sex between 2019 and 2021.

**Figure 5 ijerph-20-05315-f005:**
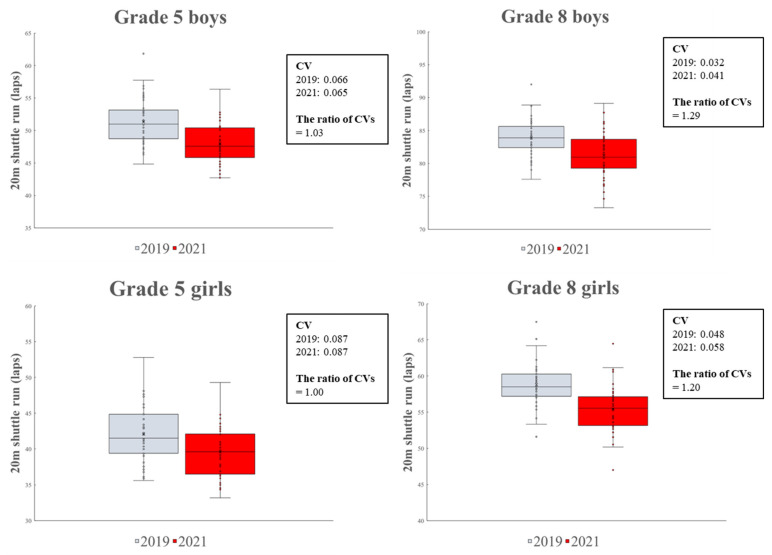
Changes in variability of CRF at prefecture levels between 2019 and 2021.

**Figure 6 ijerph-20-05315-f006:**
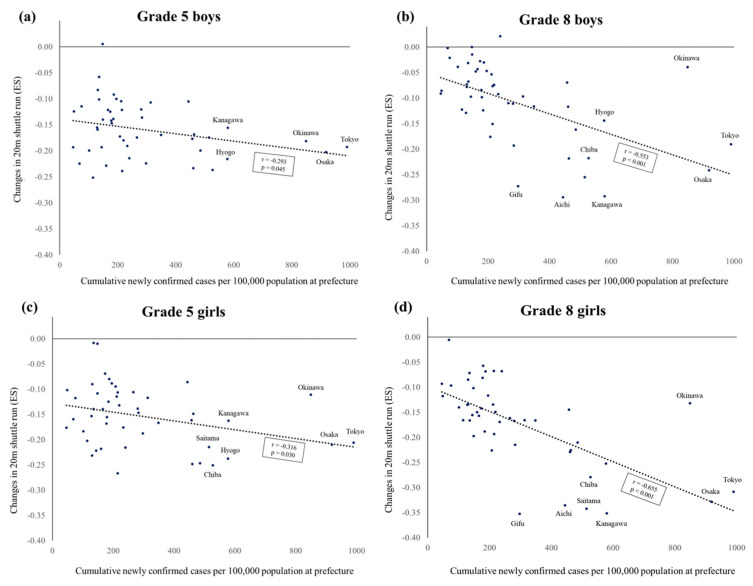
Association between the cumulative confirmed cases and changes in CRF at prefecture levels. (**a**) grade 5 boys, (**b**) grade 8 boys, (**c**) grade 5 girls, (**d**) grade 8 girls.

**Table 1 ijerph-20-05315-t001:** The 20 m shuttle run performances and their corresponding exercise times in 2019 and 2021.

	2019			2021		
	n	Mean	SD	n	Mean	SD
**20m shuttle run (laps)**						
Grade 5 boys (aged 10–11 years)	522,728	50.3	21.1	469,454	46.9	21.0
Grade 5 girls (aged 10–11 years)	504,769	40.8	16.5	449,486	38.2	16.1
Grade 8 boys (aged 13–14 years)	316,460	83.1	24.6	321,027	79.5	25.3
Grade 8 girls (aged 13–14 years)	303,414	58.0	19.9	302,202	53.9	19.8
**Exercise time (min/day)**						
Grade 5 boys (aged 10–11 years)	-	79.5	-	-	74.1	-
Grade 5 girls (aged 10–11 years)	-	49.8	-	-	47.4	-
Grade 8 boys (aged 13–14 years)	-	116.8	-	-	101.4	-
Grade 8 girls (aged 13–14 years)	-	85.1	-	-	72.3	-
**Exercise time in organized sports club (min/day)**						
Grade 8 boys (aged 13–14 years)	-	114.5	-	-	94.0	-
Grade 8 girls (aged 13–14 years)	-	115.4	-	-	92.3	-

**Table 2 ijerph-20-05315-t002:** Association among change in exercise time, change in CRF, and the number of confirmed cases.

Grade	Sex	Items	Change in CRF	The Number of Confirmed Cases (as on 30 April 2021)
			Pearson Correlation Coefficients (r)	
Grade 5	Boys	Changes in exercise time (Δ2021–2019)	0.177	−0.27
	Girls	Changes in exercise time (Δ2021–2019)	−0.075	−0.113
Grade 8	Boys	Changes in exercise time (Δ2021–2019)	0.440 **	−0.615 **
		Changes in exercise time in organized sport club (Δ2021–2019)	0.461 **	−0.691 **
	Girls	Changes in exercise time (Δ2021–2019)	0.510 **	−0.621 **
		Changes in exercise time in organized sport club (Δ2021–2019)	0.567 **	−0.704 **

** *p* < 0.05.

## Data Availability

The datasets analyzed in this study are available from the corresponding author on reasonable request.

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
