# Peer review of "Geographical Disparity in Cardiorespiratory Fitness among 3,189,540 Japanese Children and Adolescents before and during the Coronavirus 2019 Pandemic: An Ecological Study"

_ijerph, 2023, doi:10.3390/ijerph20075315_

Round 1

Reviewer 1 Report

This study is very well structured and explained. The only concern that i have and only question to the authors is: is this a secondary analysis of the data, since they note that details about the survey are published previously (line 16)? If so, I suggest adding a secondary analysis in the work's title. Overall, congrats to the authors on conducting the research.

Author Response

Comment 1: This study is very well structured and explained. The only concern that i have and only question to the authors is: is this a secondary analysis of the data, since they note that details about the survey are published previously (line 16)? If so, I suggest adding a secondary analysis in the work's title. Overall, congrats to the authors on conducting the research.

Response 1: Thank you for your positive feedback. We greatly appreciate the time and effort you have dedicated to providing this insightful feedback that will help us strengthen our paper. As you pointed out, this study did use publicly available PF data (i.e., secondary analyses). Therefore, I added this information to the abstract for better clarity (Page 1).

Reviewer 2 Report

An article titled Geographical Disparity in Cardiorespiratory Fitness Among 2,3,189,540 Japanese Children and Adolescents Before and During the 3rd Coronavirus Pandemic 2019: An Ecological Study addresses a very important public health issue in the field of physical fitness. It is particularly important to undertake research on cardiorespiratory fitness in the creative approach of creative analysis - revealing the state of the studied population before and after the pandemic (COVID-19).

The indicated global research area is a new experience requiring interdisciplinary and intercultural research. The human community expects the deepest reflection describing, ordering and diagnosing the causes and effects related to the pandemic event. Any scientific activity in this area seems to be necessary and necessary for the good of public health - especially the youngest.

A particularly valuable achievement of the research team presenting the article is the discovery that there are significant links between the accumulation of COVID-19 in prefectures, which suggests that it is important to implement activities that take into account the age and geographical location of the study population in order to strengthen the assistance provided to Japanese children and adolescents in regaining the pre-pandemic PF level. An equally important finding is the confirmation of an important relationship between the increasing geographical disparities in CRF and their impact on the health status of the study population.

Despite the undertaken - and justified - self-criticism regarding the research method and the urgent need to better correlate it with the studied population in accordance with its endogenous characteristics, the article is, in my opinion, of exceptional value, which I recommend for dissemination.

However, the article requires some corrections - elaboration and detailing - especially in the methodological layer of the research. With the above in mind, I would like to ask the authors of the article to explicitly rectify in part:

- graphical presentation of test results and a description of their statistical significance.
- readability of the article and its transparency from the perspective of the reader.

Author Response

Comment 1: An article titled Geographical Disparity in Cardiorespiratory Fitness Among 2,3,189,540 Japanese Children and Adolescents Before and During the 3rd Coronavirus Pandemic 2019: An Ecological Study addresses a very important public health issue in the field of physical fitness. It is particularly important to undertake research on cardiorespiratory fitness in the creative approach of creative analysis - revealing the state of the studied population before and after the pandemic (COVID-19).

The indicated global research area is a new experience requiring interdisciplinary and intercultural research. The human community expects the deepest reflection describing, ordering and diagnosing the causes and effects related to the pandemic event. Any scientific activity in this area seems to be necessary and necessary for the good of public health - especially the youngest.

A particularly valuable achievement of the research team presenting the article is the discovery that there are significant links between the accumulation of COVID-19 in prefectures, which suggests that it is important to implement activities that take into account the age and geographical location of the study population in order to strengthen the assistance provided to Japanese children and adolescents in regaining the pre-pandemic PF level. An equally important finding is the confirmation of an important relationship between the increasing geographical disparities in CRF and their impact on the health status of the study population.

Despite the undertaken - and justified - self-criticism regarding the research method and the urgent need to better correlate it with the studied population in accordance with its endogenous characteristics, the article is, in my opinion, of exceptional value, which I recommend for dissemination.

However, the article requires some corrections - elaboration and detailing - especially in the methodological layer of the research. With the above in mind, I would like to ask the authors of the article to explicitly rectify in part:

Response 1: Thank you for your positive feedback. We greatly appreciate the time and effort you have dedicated to providing this insightful feedback that will help us strengthen our paper. In response to your suggestions, we have added more detailed information to the methodology section of our revised manuscript (Lines 68-109).

Comment 2: - graphical presentation of test results and a description of their statistical significance.

Response 2: Thanks for your comment. However, it is unclear which results you are referring to because all of the results (except Table 2) were presented graphically (note: I have added a new table (Table 1) to the revised manuscript based on suggestion of Reviewer 3).

Regarding the description of statistical significance, it was not possible to perform statistical tests for Figures 1 and 2 because these figures only showed descriptive characteristics. For Figure 4 and Table 2, the statistical tests have already been performed (i.e., Pearson correlation) and the statistical significance has been presented. Moreover, the purpose of the Figure 3 was to demonstrate changes in the variability of CRF. Here, the ratios of CVs were estimated based on previous studies (references 19 and 20); this did not require any statistical analyses. If any further changes are required, it would be great if you specify these in greater detail so that these statistical tests can be performed if necessary.

Comment 3: readability of the article and its transparency from the perspective of the reader.

Results 3: Thank you for your comment. I have added more detailed information to describe the methodology in the revised manuscript (Lines 68-109). I hope that these additions have improved the clarity of the paper.

Reviewer 3 Report

Thank you for the opportunity to review this ecological study, which reports the associations between the extent of Covid-19 infection and changes in cardiorespiratory fitness among Japanese children and adolescents.

The topic of the study is relevant, and the results add new information about the effects of Covid-19 pandemic on fitness and well-being of children and adolescents.

The study is based on a nationally representative data on children’s and adolescents’ cardiorespiratory fitness. The text is well-written and easy to follow. The introduction provides a sound rationale for the study and the methods are clearly described. Conclusions are in line with the findings.

I found few points that might be useful to clarify further:

-          The results section could be strengthened by starting it with descriptive results on the background characteristics: participants age and sex, the fitness level as well as self-reported exercise time and organized sport participation before and after the pandemic (on a group level and /or according to prefecture).

-          The results could be interpreted in more detail and may be the order of presenting could be revised for example by presenting the main results of each figure or table at the beginning of each paragraph and referring to the figure/table after that. Now each paragraph begins with the reference to the figure/table.

-          The methodological texts under the figures could be removed to the methods section (most of them already are there).

-          Table 1. Please indicate 1) the direction of changes in exercise times, 2) the r in the column titles and 3) which associations are presented. It seems that the change in CRF column presents the association (r) between the exercise-item and this change, and the number of confirmed cases column presents the association (r) between the item and the number. Please specify.

Author Response

Reviewer 3

Comment 1: The results section could be strengthened by starting it with descriptive results on the background characteristics: participants age and sex, the fitness level as well as self-reported exercise time and organized sport participation before and after the pandemic (on a group level and /or according to prefecture).

Response 1: Thank you for this constructive suggestion. Accordingly, I have added a new table (Table 1) to the revised manuscript. This table describes the background characteristics of the participants, including age, PF, and exercise time before and during the pandemic (Table 1).

Comment 2: The results could be interpreted in more detail and may be the order of presenting could be revised for example by presenting the main results of each figure or table at the beginning of each paragraph and referring to the figure/table after that. Now each paragraph begins with the reference to the figure/table.

Response 2: Thank you for your suggestion. I have included some additional information in the text. Additionally, I have changed the order in which the content is presented, which will aid the readers’ understanding of our results (Pages 4–11).

Comment 3: The methodological texts under the figures could be removed to the methods section (most of them already are there).

Response 3: As you suggested, I have removed figure legends from the text, as these were already mentioned in the methodology section.

Comment 4: Table 1. Please indicate 1) the direction of changes in exercise times, 2) the r in the column titles and 3) which associations are presented. It seems that the change in CRF column presents the association (r) between the exercise-item and this change, and the number of confirmed cases column presents the association (r) between the item and the number. Please specify.

Response 4: Thanks for your suggestion. I have modified the table (this is designated as Table 2 in the revised manuscript) based on your suggestion (Table 2).

Reviewer 4 Report

Tetsuhiro Kidokoro presented the article "Geographical disparity in cardiorespiratory fitness among 3,189,540 Japanese children and adolescents before and during the coronavirus 2019 pandemic: an ecological study".

Dear author, congratulations on your article. The subject is very interesting. However, some changes are in order:

Methods: it is necessary to clarify the survey. Naming previous paper as a refference is not acceptable.

Line 67-68: Improve sentence  "The participation rate in the PF survey at school level was almost perfect." with some more scientific vocabulary.

Line 100-101: As abovementioned, it is not appropriate to simply note "described elsewhere". This paper is a work for itself, so it is necessary that everything is clarified.

Author Response

Reviewer 4

Comment 1: Tetsuhiro Kidokoro presented the article "Geographical disparity in cardiorespiratory fitness among 3,189,540 Japanese children and adolescents before and during the coronavirus 2019 pandemic: an ecological study". Dear author, congratulations on your article. The subject is very interesting. However, some changes are in order:

Response 1: Thank you for your positive feedback. We greatly appreciate the time and effort you have dedicated to providing this insightful feedback that will help us to strengthen our paper.

Comment 2: Methods: it is necessary to clarify the survey. Naming previous paper as a refference is not acceptable.

Response 2: Thanks for your constructive suggestion. In the revised manuscript, I have included more detail information in the methodology section (Lines 68-109).

Comment 3: Line 67-68: Improve sentence "The participation rate in the PF survey at school level was almost perfect." with some more scientific vocabulary.

Response 3: As this sentence was unclear, it has been removed from the revised manuscript. Instead, I have shown the participation rates for the PF survey in 2019 and 2021 for all age- and sex-groups (Lines 68-73).

Comment 4: Line 100-101: As abovementioned, it is not appropriate to simply note "described elsewhere". This paper is a work for itself, so it is necessary that everything is clarified.

Response 4: As you suggested, I have added the specific methodology used to calculate the CV ratio in the revised manuscript (Lines 108-109).
